# Trends and Challenges in the Mental Health of University Students with Disabilities: A Systematic Review

**DOI:** 10.3390/bs14020111

**Published:** 2024-02-02

**Authors:** Patricia Solís García, Sara Real Castelao, Alejandra Barreiro-Collazo

**Affiliations:** 1Faculty of Education, Universidad Internacional de La Rioja, 26006 Logroño, Spain; patricia.solis@unir.net (P.S.G.); alejandra.barreiro@unir.net (A.B.-C.); 2Faculty of Education, Universidad de León, 24071 León, Spain

**Keywords:** mental health, higher education, university students with disabilities

## Abstract

This systematic review examines mental health in university students with disabilities, focusing on increasing prevalence and associated challenges. Following the PRISMA protocol for study selection and analysis, it aims to analyze problem prevalence and risk factors, explore support strategies and available resources, and identify gaps and areas for improvement in care and access to mental health services for disabled university students. This review includes 16 articles that met the established criteria. The findings reveal higher mental health issue prevalence in these students compared to non-disabled peers, highlighting the need for specific, tailored interventions. Improvement areas in general inclusion measures to prevent high-risk situations and intervention responses to discomfort and existing mental health problems are discussed. The importance of a holistic approach to ensure their overall well-being and academic success in an inclusive educational environment is emphasized.

## 1. Introduction

The improvement in educational systems has led to greater access to higher education for the population. This increased access is particularly notable among students with disabilities, whose presence has increased at a higher percentage than in the general population over the last two decades [1,2]. Thus, higher education increasingly includes students with a wide range of disabilities (e.g., intellectual, motor, sensory, Autism Spectrum Disorder, etc.) due to the success of previous stages marked by legal obligations towards the development of inclusive and equitable education, with programs, measures, resources, and support to achieve the highest levels of equity in access, retention, and graduation [3]. Therefore, institutions of higher education should not only consider academic and learning aspects but also maximize the holistic development of students, taking into account both physical and mental health factors.

Universities, as fundamental pillars for the advancement of knowledge and societal enhancement, play a crucial role in promoting the inclusion of all students [4]. This responsibility becomes particularly relevant in the context of educational inclusion for people with disabilities, an objective that requires significant changes throughout the educational system to achieve opportunity equity [5]. This shift towards a more inclusive education has allowed expanded access to groups that have historically been excluded. However, despite these improvements, students with disabilities face unique and additional challenges in adapting and meeting the demands of university life [6].

Understanding mental health in the university setting is a complex issue involving an intersection of biological, psychological, social, and economic factors [7]. All university students face significant levels of stress, anxiety, and emotional issues, a reality supported by various research studies [8,9], which have documented high rates of somatic and psychological complaints in this group. Consistent with these concerns, more recent studies [10] suggest an increase in the prevalence of mental health problems among university populations over the last decade. Moreover, recent research [11] encompassing university students from eight countries found that 34% reported experiencing at least one situation related to a mental disorder in their lifetime. Other studies emphasize the effects of distance education on the mental health of students [12]. These data highlight the magnitude of the impact of mental disorders on the student population, with over half of the respondents reporting feelings of hopelessness and nearly 90% facing overwhelming situations in the past 12 months [9]. This situation underscores the urgency for academic institutions to implement multidisciplinary measures to address this challenge and ensure the mental well-being of their students, emphasizing the need to evaluate how this context impacts specific populations, such as students with disabilities.

University students with disabilities face significant challenges in their mental health. Studies conducted following the crisis caused by COVID-19 [13,14] indicate a higher risk among individuals with physical disabilities [2]. However, the limited number of existing studies in this area are based on small and non-comparable sample sizes of university students with disabilities [15,16]. Studies comparing the mental health needs of students with disabilities to those without disabilities indicate a significantly higher prevalence of depression, anxiety, non-suicidal self-injury, and suicidal risk among students with disabilities [17], as well as higher rates of suicidal ideation and suicide attempts [16,18]. Students with disabilities face a higher risk of stigma; those who may be perceived as dangerous (e.g., psychopathy) were more stigmatized [19]. Apart from data derived from the general university context, it has been highlighted that the lack of adequate adaptations has been associated with an increased risk of depressive symptoms [20].

Furthermore, the variation in requests for accommodations based on disability categories can reveal barriers and vulnerabilities within specific subpopulations of students with disabilities [21]. Approximately half of university students who reported disabilities requested accommodations. This proportion was lower among those with cognitive disabilities, such as attention deficit hyperactivity disorder, learning disabilities, and psychological disabilities. This group of students also faces stigmas and misconceptions about their capability in academic performance [21]. These findings underscore the importance of understanding differences in the needs and experiences of university students with disabilities to provide a more inclusive educational environment and support their mental well-being.

Based on the described data, conducting a systematic review in this field is of crucial importance to deepen the understanding and approach to the mental health of university students with disabilities. Given the complexity of this issue, involving a blend of interrelated factors from biological to social aspects, previous research [11,17] has emphasized the urgency of a more comprehensive evaluation. Thus, this review will identify patterns, challenges, and specific needs faced by these students, aiming to understand the magnitude of mental disorders in this population. Additionally, by understanding variations in accommodation requests and differences among various disabilities, more precise and effective strategies can be designed to offer an inclusive educational environment and comprehensive support for their psychological well-being from health promotion and prevention perspectives. Similarly, this review aligns with the Global Goals of the 2030 Agenda for Sustainable Development, as this study aims to understand the mental health of university students with disabilities, contributing to the promotion of good health and well-being. Likewise, by examining the challenges faced by students with disabilities and emphasizing the need for inclusive educational environments, this study aligns with the intention to provide quality education that is equitable and inclusive.

The overall objective of this study is to conduct a comprehensive systematic review of the state of mental health among university students with disabilities, aiming to understand their current situation, identify trends, and highlight the challenges they face, prompting an appropriate response from academic institutions. Similarly, the following specific objectives are pursued:Analyze the prevalence and associated risk factors of mental health problems among university students with disabilities.Explore the support strategies and available resources to enhance the mental health of this student group.Identify gaps and areas for improvement in the provision of care and access to mental health services for university students with disabilities.

## 2. Materials and Methods

This systematic review follows the guidelines outlined in the Preferred Reporting Items for Systematic Review and Meta-Analysis (PRISMA) framework, as recommended [22,23]. However, since it does not include a health outcome with direct relevance to human health, it has not been registered in a systematic review registry like PROSPERO. Likewise, due to the thematic variability found, it was not possible to conduct meta-analyses.

### 2.1. Phenomenon of Interest

The research questions guiding this review are as follows: (1) What is the prevalence of mental health problems among university students with disabilities? (2) What are the associated risk factors for mental health problems in university students with disabilities? Are there specific factors that increase the vulnerability of this group? (3) What are the main challenges faced by university students with disabilities concerning their mental health? Are there significant barriers to accessing mental health services?

### 2.2. Study Selection

#### 2.2.1. Inclusion and Exclusion Criteria

The PRISMA protocol [22] was employed to conduct the search and evaluation of the found studies. The inclusion criteria were as follows: (1) full articles published in peer-reviewed journals in English or Spanish from 2018 to July 2023 and (2) empirical articles (quantitative or qualitative) related to the mental health of university students with disabilities. The last five years were selected to obtain a current perspective on the subject. The following types of articles were excluded: (1) those not peer-reviewed, such as book chapters, theses, reviews, and conference abstracts; (2) those addressing different topics or not specifically including students with disabilities in their samples.

#### 2.2.2. Information Sources and Search Strategy

This systematic review focused on sourcing complete works from primary publications available in the Web of Science (WOS), SCOPUS, and PsycINFO databases. WOS and SCOPUS cover a wide range of disciplines, proving useful when seeking an interdisciplinary perspective or a broader view of research in a particular area. On the other hand, PsycINFO specializes in psychology and related disciplines, making it potentially more comprehensive and specific in our search. This facilitated access to a relevant set of databases for the subject matter at hand.

To locate articles, the search utilized keywords from the APA Thesaurus, using the keywords “mental health” AND “university students” AND “disability”.

#### 2.2.3. Data Screening and Extraction

The PRISMA flow diagram [23], depicted in Figure 1, was utilized for article selection. Initially, employing the list of descriptor terms from the search yielded a total of 401 articles. Subsequently, duplicate articles were removed, and the three researchers assessed 340 articles to determine their relevance to the topic of interest (i.e., addressing variables related to mental health or emotional well-being in university students with disabilities). Of these, 71 were identified as theoretical studies lacking empirical data, and 144 did not align with the subject matter. Moreover, out of the 125 articles attempted for retrieval, access to 41 articles was unsuccessful. Ultimately, following an in-depth reading of the remaining 85 articles by two authors, 68 were excluded based on exclusion criteria. Meanwhile, the remaining author supervised their colleagues’ work to ensure accuracy. In this way, the third researcher provided supervision to facilitate a consensus decision in case of disagreement regarding the relevance of an article. The data extraction table encompassed the study’s objectives, participants, study design, and key discoveries. Detailed records of decisions regarding article inclusion were maintained for comprehensive tracking purposes; for this purpose, the Parsifal tool was used, which is an online tool designed to support researchers to perform systematic literature reviews within the context of Software engineering.

Given the diverse range of outcome measures, performing a meta-analysis was unfeasible. Instead, a narrative synthesis method was adopted. This involved organizing the findings from the studies by grouping them according to the involvement of mental health conditions in students with disabilities in university settings. Each category’s findings were thoroughly analyzed to identify common themes and patterns. Moreover, a comprehensive integration of the findings from all studies was conducted, considering variations in sample characteristics, methodological design, and quality. This process aimed to highlight significant trends across the gathered data.

Regarding the limitations of the evidence included in this review, the focus on articles published in peer-reviewed journals may introduce publication bias. Unpublished studies, grey literature, or studies in languages other than English or Spanish may not be included. The inclusion criteria restrict the review to articles published from 2018 to July 2023. While this ensures a recent perspective, it may exclude relevant earlier studies that could contribute to a more comprehensive understanding of the topic. Then, again, we encountered access limitations; the unsuccessful retrieval of 41 articles due to access issues may introduce a potential source of bias if those articles contained important insights relevant to the review. The relevance assessment conducted by the three researchers involves subjective judgment, and, despite efforts to maintain consistency through supervision, variations in interpretation may still exist. Finally, the decision not to perform a meta-analysis limits the ability to quantify the overall effect size and assess the statistical significance of the findings across studies.

## 3. Results

### 3.1. Study Characteristics and Participants

This study covers 16 articles involving a total sample of 138,547 students, out of which 16,327 presented with disabilities. The difference arises because 6 out of the 16 articles conducted studies comparing university students with disabilities to their non-disabled counterparts. The sample sizes across the studies varied significantly, ranging from 25 to 93,348 (6382 in samples of students with disabilities). Seventy-five per cent (12/16) of the studies employed qualitative methodologies, adopting non-experimental, descriptive, cross-sectional, and quasi-experimental designs. Validated measures, such as the Multidimensional Body-Self Relations Questionnaire (MBSRQ), Coping Styles Questionnaire (CQS), Symptom Checklist 90 (SCL-90), Self-Stigma of Disability Scale (SSDS), Self-Acceptance Questionnaire (SAQ), and the University of California, Los Angeles (UCLA) Loneliness Scale were utilized. Two studies (12.5%) employed mixed methods, while two others adopted a qualitative approach based on semi-structured interviews. Regarding the distribution over the last five years, we can see that the year 2021 has the highest number of publications (*n* = 6). However, only two have examined the impact of the epidemic on the mental health of students with disabilities (in the years 2022 and 2023).

### 3.2. Areas of Focus

According to the described criteria, the selected studies have been grouped into four areas based on the themes and objectives established in the reviewed studies. The analysis conducted is documented in Table A1, Table A2, Table A3 and Table A4 of Appendix A.

In the first thematic area (Table 1), three articles focus on the overall mental health of university students with disabilities.

The first study [16] in this group stands out due to its large sample size of 93,348, including 6382 students with disabilities. The study’s results revealed that students with disabilities exhibited significantly higher rates of mental health issues (67% compared to 45% for students without disabilities (*p* < 0.0001)), with a significantly higher prevalence of depression, anxiety, non-suicidal self-injury, and a threefold likelihood of suicide attempts. They were also more likely to use mental health services compared to their counterparts without disabilities. Likewise, the prevalence of prescription medication use was twice as high among students with disabilities who reported at least one mental health issue compared to students without disabilities. Other authors have found similar results, highlighting a higher prevalence of mental health problems in this group; experiencing microaggressions can provoke feelings of embarrassment, anger, and frustration [1].

One of these studies [24] focused on discrimination experiences, significantly predicting higher symptoms of anxiety and depression, poorer academic self-image, and lower satisfaction with grades. Another study [1] investigated the mediating role of coping strategies between body image assessment and mental health in Chinese university students with three types of disabilities: physical, visual, and auditory. Their results indicated that the mental health status of the three groups of students with disabilities was significantly lower than the Chinese norm, especially among those with physical disabilities. Furthermore, they found that coping strategies played a central role, and their specific role varied based on the type of disability in the connection between body image assessment and student mental health.

Specifically, students with physical disabilities scored higher on positive coping strategies compared to students with visual and auditory disabilities. However, despite the more frequent use of positive strategies, they had an especially unfavorable mental health status. On the other hand, a better body image was associated with better mental health in physical and auditory disabilities. However, in visual disabilities, body image did not show a significant relationship.

Group 2 (Table 2), comprising 5 studies, addresses programs, measures, and actions for university students with disabilities, providing a detailed insight into different intervention approaches and their outcomes.

Some authors [25] conducted a multicenter randomized clinical trial to assess the effectiveness of the “ACCESS—Accessing Campus Connections and Empowering Student Success” program in university students with Attention-Deficit/Hyperactivity Disorder (ADHD). While no improvements were observed in depression and anxiety, the analysis suggests that ACCESS participants were less likely to experience a worsening of depression/anxiety symptoms, supporting the viability of this program as a treatment for young adults with ADHD in university settings. Similarly, significant decreases were shown in overall ADHD symptomatology, general deficits in executive functioning, as well as significant growth in their knowledge of ADHD, use of behavioral strategies, and use of disability services.

Others [26] took a qualitative approach to understand the experiences of autistic university students with Disability Services Offices (DSOs) in higher education institutions in the United States. The students expressed mixed experiences with the support provided by DSOs. On the one hand, they appreciated academic accommodations as they increased their focus and academic competence and improved their emotional state, along with non-academic support that increased their sense of belonging and opportunities to meet others with disabilities. However, they pointed out areas for improvement, such as effective communication between professors and DSOs, training on specific ASD-related knowledge, and the accessibility of staff or appointment scheduling length.

Article three [21] provides detailed insight into academic accommodation requests among second-year medical students across different disability categories. Data were collected from a national online survey where, out of 27,009 students, 9.0% identified themselves as individuals with disabilities. Among these, cognitive disabilities were the most common (77.3%), followed by chronic health conditions (15.7%) and motor/sensory disabilities (7.0%). Out of a total of 1108 accommodation requests, 71.7% were made by students with cognitive disabilities, 19.8% by those with chronic health conditions, and 8.5% by students with motor/sensory disabilities. A significant finding was that 19.0% of the students who reported needing accommodations did not request them, with this non-request proportion being higher among students with cognitive disabilities compared to other disability categories. This suggests potential barriers or stigmas associated with cognitive disabilities that might influence the lower frequency of accommodation requests in this specific group. The results underscore the importance of understanding the needs and obstacles faced by students with cognitive disabilities in university settings, especially concerning misconceptions and stigma surrounding their ability to succeed academically.

On the other hand, ref. [27] reports that almost a third of the students did not use some or all of the accommodations offered by the university either because they felt they were not helpful or due to external barriers, such as unresponsive professors or tight application deadlines. Additionally, there were statistically significant positive relationships between general comfort disclosing and comfort disclosing to specific interaction partners (faculty, staff, students). Finally, peers were reportedly significantly less respectful and positive when students disclosed accommodation needs, and students felt less comfortable discussing accommodation needs with their peers.

Finally, ref. [28] aimed to characterize the population of individuals with ASD in the university. Their findings underscore the fact that students with ASD reported similar rates of ADHD, learning disabilities, and psychological comorbidities compared to clinical referral samples. Additionally, certain factors, such as gender and comorbid diagnoses, may influence the psychological well-being and academic disengagement of these students.

In summary, these studies provide diverse insight into support programs and experiences of students with different disabilities in university settings, highlighting both positive aspects and critical areas that require attention and improvement.

Group 3 (Table 3) brings together six investigations that examine the prevalence, experiences, and challenges in physical and mental health among university students with various disabilities.

Firstly, ref. [29] focused on the prevalence of tobacco use among college students with disabilities, noting a higher likelihood of current tobacco consumption and nicotine dependence in this group, with an 86% higher likelihood of meeting the criteria for nicotine use disorder. This underscores the need for specific services to address the risk of smoking among students with disabilities. On the other hand, ref. [30] explored sexual victimization in college students with disabilities, identifying a higher probability of victimization both pre-college and during college compared to students without disabilities. This study highlights the need for relevant support services and the evaluation of a university’s culture as key points in addressing sexual violence.

By contrast, other authors [31] focused on loneliness among college students with visual impairments, finding significant associations between visual impairment, friendships, and loneliness. This study observed a significant correlation between self-stigma, self-acceptance, and loneliness among students with visual impairments. That is to say, their sense of self-stigma is associated with lower self-acceptance and a higher level of loneliness, and, conversely, lower self-acceptance is accompanied by a higher sense of loneliness. Given this relationship, the importance of interventions that improve self-acceptance to mitigate self-stigma and loneliness in this group is emphasized.

When comparing the experiences of students with ASD with other disabilities and neurotypical students, they found [16] that both groups of students with disabilities reported worse outcomes in various areas, both in physical and psychological health. Specifically, they perform worse academically, have lower feelings of belonging, lower-quality social relationships, higher rates of physical and verbal harassment, are more likely to report feeling different from other students, have worse health, and have higher alcohol consumption rates, indicating challenges related to stigma and social rejection associated with disability in general. On the other hand, [2], investigating differences in perceived social support, depressive symptoms, and well-being between students with physical disabilities living in general accommodations versus communities of people with disabilities highlighted the importance of social support. It suggested implementing programs to foster community among students with physical disabilities.

Finally, ref. [32] explored differences between first-year students with ASD and students with learning disabilities. They identified similarities and differences in characteristics and behaviors. Overall, the characteristics of students with ASD are similar to those with LD. However, people with ASD show a higher level of intellectual self-confidence but fewer interactions with professors and students. Those with LD score higher in risk-taking and engage more in group projects. Therefore, it is important to highlight the importance of understanding the specific needs of these groups to provide a supportive environment in university. In summary, these studies highlight the importance of understanding the unique experiences, challenges, and needs of college students with disabilities, emphasizing the need for specific services and supportive environments to ensure their academic success and physical and mental well-being.

In conclusion, Group 4 (Table 4) comprises two research studies on the impact of the pandemic on students with disabilities, highlighting the difficulties encountered and the areas that need specific attention in situations such as those experienced during the COVID-19 crisis.

Firstly, ref. [12] focused on the psychological well-being of students with different somatic health conditions before and after the COVID-19 pandemic. The results indicated a significant decline in psychological well-being, especially among students with chronic illnesses and short-term rare disabilities. This group experienced a marked decrease in autonomy, environmental management, and personal growth, with a prevalence of negative effects and a higher likelihood of developing anxiety and depression. Thus, the negative impact of the pandemic on their psychosocial well-being is emphasized.

On the other hand, ref. [13] explored the experiences of students with disabilities during the abrupt shift to remote education due to the pandemic. They observed that most elements related to access to services, instruction, and academic policies did not show improvements. Aspects associated with mental health, motivation to learn, and connections with peers were perceived as worse than in the previous semester. These findings underscore the urgent need to enhance access to services and promote instructions with a Universal Design for Learning (UDL) framework. Moreover, providing greater support in terms of mental health and motivation is necessary to address the needs of students with disabilities in a remote education setting. Both studies highlight the specific challenges faced by students with disabilities during the pandemic, emphasizing the importance of addressing their particular needs to ensure their psychological well-being, access to services, and effective instruction, especially in the remote educational environment.

## 4. Discussion

The overarching goal of this systematic review has been to explore the state of mental health among university students with disabilities. The purpose of this research included comprehensively understanding their current situation, identifying patterns, and highlighting the challenges they face within the university context. Throughout this study, the aim has been to gather and analyze a wide range of research to provide comprehensive insight into the experiences, obstacles, and specific needs of this student group concerning their overall well-being.

The primary specific objective was centered on analyzing the prevalence and risk factors associated with mental health issues among university students with disabilities. Studies compiled through the PRISMA flowchart consistently highlight a higher prevalence of mental health problems among students with disability issues (the odds of meeting the criteria for any mental health problem were 2.5 times greater among students with disabilities (*p* < 0.0001, 95% CI: 2.4–2.6) [17], showing increased risks related to emotional, behavioral, and academic needs, as well as fewer coping strategies compared to their non-disabled peers [1,17]. Additionally, a tendency towards a greater utilization of mental health services by these students has been identified [17]. Moreover, the results reveal a diversity of crucial aspects to understand the prevalence and factors associated with mental health problems in these students. This research examined multiple areas and factors, including tobacco use among students with disabilities, comorbidity with other disabilities or illnesses, gender, or the students’ self-conception, among others [29]. Moreover, this research highlights the higher likelihood of tobacco use and nicotine dependence within this group, emphasizing the need for specific services to address the risk of smoking, such as disability support services providing students with accessible information about mental health centers and smoking cessation programs. On the other hand, ref. [30] emphasizes the higher prevalence of sexual victimization among students with disabilities, stressing the importance of implementing support services and conducting an evaluation of the university’s culture regarding sexual abuse. Furthermore, studies [31] identified loneliness (the loneliness scores of visually challenged college students were at a high level (mean ¼ 44.97, SD ¼ 9.35) and were also higher than those of other groups, such as other students without disabilities), discrimination, and exclusion as significant factors among students with visual disabilities, emphasizing the importance of self-acceptance in mitigating self-stigma and loneliness. Likewise, all comparative studies found in the review show a higher prevalence of mental health problems and worse indicators in the population of students with disabilities compared to their non-disabled counterparts [12,17,21,29,30]. For example, when comparing conditionally healthy students with students with chronic diseases with rare and short-term disabilities, significant differences were found on all scales of psychological well-being. The scores on the “Autonomy” (*p* = 0.015), “Environment Management” (*p* = 0.046), and “Personal Growth” (*p* = 0.009 & *p* = 0.003) scales for the second group dropped sharply [12]. These differences are not only observed in the neurotypical population but also among different types of disabilities [16] (students on the spectrum and students with other disabilities reported poorer health than neurotypical students on most indicators. Their health disadvantage extended to self-reported physical health [F(2, 3070) = 39.59, *p* = 0.00], self-reported mental health [F(2, 3068) = 132.97, *p* = 0.00], depression [F(2, 3065) = 87.08, *p* = 0.00], and anxiety [F(2, 3059) = 16.69, *p* = 0.00]), where students on the ASD and others with disabilities exhibit common challenges associated with academic performance, stigma, social rejection, and health (with the group of students with ASD experiencing greater difficulties in these areas). In terms of prevalence among specific subgroups, ref. [28] provides comprehensive insight into students with ASD in university settings, highlighting similarities in the prevalence of certain conditions and the influence of factors, such as gender and comorbid diagnoses, on psychological and academic well-being [32].

These collective findings underscore the need for specific strategies and suitable support environments with a multifaceted approach to ensure the mental and academic well-being of university students with disabilities. It is crucial to consider their unique experiences and particular challenges, with a special emphasis on crises [12] or discrimination-related issues [16,24,30]. Therefore, research on this topic confirms the higher prevalence within this group compared to the general population, as previously indicated by studies [8,9,11,18]. It also highlights personal and contextual factors that increase the risk of mental health problems and hinder the implementation of treatments.

Regarding the second objective, this review explored support strategies and available resources to enhance the mental health of this student group. The diverse support programs and resources for university students with disabilities present a complex and varied landscape in their outcomes [25]. It stresses the potential effectiveness of programs like ACCESS for students with ADHD, as it reduces general symptomatology, enhances executive functions, and mitigates the worsening of depressive and anxiety symptoms. However, not all aspects showed significant improvements. On the other hand, the experiences of students with ASD within disability support offices, as analyzed [27], highlight critical areas for improvement. Lastly, ref. [2] highlights the effectiveness of mutual support among physically disabled individuals within the university to counter feelings of isolation and loneliness. These findings represent examples of best practices that can be applied to other contexts but require follow-up evaluations and further evidence. The descriptive and cross-sectional nature of the studies indicates the nascent understanding in this area.

Regarding the third and final objective, the comprehensive research on the attention and access to mental health services for university students with disabilities reveals multiple areas for improvement and significant gaps. This research finds that the higher prevalence correlates with a trend towards increased use of mental health services by these students [17], although the support provided by universities is insufficient to mitigate the negative effects [24]. Furthermore, deficiencies are evident in the existing general support programs. Although interventions have been implemented, the results show varied impacts and, in some cases, contradictions in the effectiveness of these initiatives [25]. Key aspects, such as lack of specific knowledge about certain disabilities, staff accessibility, and campus culture, represent critical areas that require improvement in support services [26]. The study during the COVID-19 pandemic [13] underscores the importance of formulating support strategies and implementing programs that improve the psychological well-being of students with chronic illnesses and rare disabilities. The study on academic accommodations in medical students [21] points out potential stigma, lack of awareness about support options, misinformation, and the social and academic repercussions associated with cognitive disabilities, evidenced by the lower frequency of accommodation requests in this specific group. These findings emphasize the urgent need for more holistic approaches, specific training, and a better understanding of individual needs to overcome barriers and improve support for university students with disabilities. In the same vein, [14] highlights the pressing need for specific interventions and proactive measures for the academic and emotional improvement of individuals with disabilities.

These results underscore the urgent need for more effective and tailored interventions that address the complex intersections between disabilities, mental health, and university experiences, thereby ensuring a more inclusive environment and effective support for these students.

## 5. Conclusions

This systematic review provides comprehensive and detailed insight into the mental health of university students with disabilities. It addresses key aspects, such as prevalence, risk factors, available strategies, and resources within the university context and identifies areas for improvement and needs in their care.

The reviewed results lead to the conclusion that there is a higher prevalence of mental health issues among students with disabilities compared to their non-disabled peers, with a particular vulnerability seen in groups with cognitive disabilities. This vulnerability is not solely due to the limitations of the diagnosis itself. However, it is significantly impacted by attitudinal barriers and prejudices within the environment, affecting even the student’s self-concept. Additionally, sociodemographic factors, such as gender, reveal greater vulnerability in women, higher rates of victimization in situations of sexual abuse, and increased consumption of substances like tobacco.

This higher-risk situation lacks established treatments beyond the description of best practices within disability support services available at many universities. The findings indicate a need for improvement in general inclusion practices, including the promotion of inclusive culture and greater awareness among key stakeholders, such as teachers, regarding disability situations. There are even more significant gaps concerning mental health services in terms of access and quality, emphasizing the urgent need for more tailored and effective interventions to ensure an inclusive environment and appropriate support for these students, respecting their individual needs and the complex intersections between disability, mental health, and the university environment. Therefore, there is a clear need to delve deeper into the study of mental health among students with disabilities in higher education institutions. This comprehensive approach aims to assess, intervene, and evaluate follow-ups to establish effective interventions and prevention protocols. Only through these measures can universities truly become environments that foster the holistic development of their students, aligning with the missions and values of such institutions.

This study significantly contributes to achieving the Global Goals of the 2030 Agenda for Sustainable Development, particularly Goal 3: Good Health and Well-being and Goal 4: Quality Education. By systematically examining the mental health of university students with disabilities, this research sheds light on the prevalent challenges and higher-risk situations faced by this vulnerable group. The findings underscore the need for targeted interventions and improvements in inclusive practices within educational institutions, aligning with Goal 3’s objective to ensure mental health and well-being for all. Moreover, this study emphasizes the importance of an inclusive educational environment for students with disabilities, aligning with Goal 4’s aim to provide quality education that is inclusive and equitable. The identified gaps in mental health services underscore the urgency of tailored interventions, addressing the complex intersections between disability, mental health, and the university environment. Ultimately, this study advocates for a holistic approach to creating supportive, inclusive environments in higher education institutions, aligning with the broader principles of the 2030 Agenda for Sustainable Development.

## Figures and Tables

**Figure 1 behavsci-14-00111-f001:**
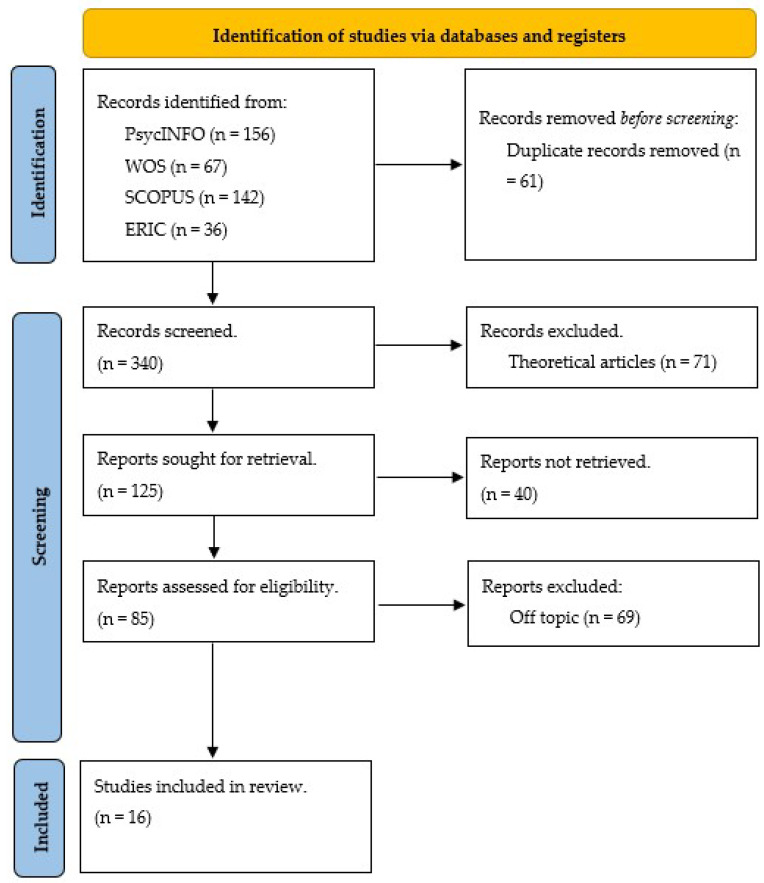
Article selection process (based on [21]).

**Table 1 behavsci-14-00111-t001:** Group 1: Mental health and university students with disabilities.

Authors	Objectives	Key Findings
Aguilar y Lipson (2021) [17]	Estimating the prevalence of mental health symptoms and help-seeking rates in a sample that includes university students with disabilities across the United States.	Students with disabilities exhibited significantly higher rates of mental health issues and were more likely to use mental health services compared to students without disabilities.
Lett et al. (2020) [24]	Examining the relationship between experienced discrimination, academic performance, and mental health outcomes in Canadian university students with disabilities.	Experiences of discrimination significantly predicted higher symptoms of anxiety and depression, poorer academic self-image, and lower satisfaction with grades.
Xu y Liu (2020) [1]	Investigating the mediating role of coping strategies between body image assessment and mental health and their variations among Chinese university students with three types of disabilities: physical, visual, and auditory.	The mental health status of the three groups of students with disabilities was significantly lower than that of the general reference population.

**Table 2 behavsci-14-00111-t002:** Group 2: Programs, measures, and support actions for university students with disabilities.

Authors	Objectives	Key Findings
Anastopoulos et al. (2021) [25]	Examine the effectiveness and feasibility of a cognitive-behavioral therapy program called “ACCESS—Accessing Campus Connections and Empowering Student Success” in university students with Attention-Deficit/Hyperactivity Disorder (ADHD).	No improvements in depression and anxiety were observed through latent growth curve analysis. RCI (reliable change indices) analyses indicated that ACCESS participants immediately showed significant decreases in general ADHD symptomatology and were less likely to report worsening symptoms of depression/anxiety.
Kim and Crowley (2021) [26]	Understand the perspectives and experiences of autistic university students with Disability Services Offices (DSOs) and the available support services from these DSOs in higher education institutions in the United States.	Students expressed both positive and negative experiences with the academic and non-academic support provided by DSOs. Staff in DSOs were perceived to lack specific knowledge about Autism Spectrum Disorder (ASD) and were often inaccessible.
Meeks et al. (2022) [21]	Identify the proportion of requests for academic accommodations in different disability categories.	Of students with disabilities, 51.8% requested accommodations (for cognitive disability, chronic health, and motor/sensory and cognitive disabilities). A total of 36.1% reported not needing accommodations, and 12.1% did not request them for reasons other than not needing them.
Smith et al. (2019) [27]	Identify specific areas of distress associated with the accommodation request process and explore individual differences related to the visibility of the condition.	Students with non-apparent disabilities related to mental health reported significantly higher distress when disclosing and more negative interactions with peers than students with apparent conditions or non-apparent learning difficulties.
Sturm and Kasari (2019) [28]	Deepen the understanding of the population of individuals with ASD to provide valuable insights to higher education institutions seeking to respond to the increasing need for support services for individuals with ASD.	University students with ASD reported overall similar rates of ADHD, learning disabilities, and comorbid psychological disorders compared to clinic-referred samples, being more similar than different from their typically developing peers. A comorbid diagnosis of ADHD was associated with higher academic detachment. Women with ASD and those with any comorbid disorder were more likely to report poorer psychological health.

**Table 3 behavsci-14-00111-t003:** Group 3: Prevalence, experiences, and challenges in physical and mental health among university students with disabilities.

Authors	Objectives	Key Findings
Casseus et al. (2022) [29]	Describe the prevalence of tobacco use in a nationally representative sample of university students with disabilities.	The prevalence of tobacco use is higher among students with disabilities compared to those without disabilities. Students with disabilities were more likely to be current tobacco users and also had higher odds of nicotine dependence.
Kirkner et al. (2022) [30]	Understand the sexual victimization of university students with disabilities in a large academic institution in the Mid-Atlantic region using an intersectional approach.	Students with disabilities showed a significantly higher likelihood of sexual victimization before arriving on campus and while in college, with much higher rates of pre-college victimization compared to students without disabilities.
Kong et al. (2021) [31]	Examine the current status of loneliness among university students with visual disabilities, along with their influencing factors, and explore the mediating role of self-acceptance between self-stigma and loneliness.	Visual disability status and relationships with friends were significantly associated with loneliness. Additionally, a significant mediating role of self-acceptance between self-stigma and loneliness was observed in students with visual disabilities.
McLeod et al. (2019) [16]	Describe the academic, social, and health experiences of university students with ASD compared to students with other disabilities and their neurotypical peers without disabilities.	There were few significant differences between students with ASD and those with other disabilities. Both groups of students reported significantly worse outcomes than neurotypical students in academic performance, social relationships and bullying, and physical and mental health.
Minotti et al. (2021) [2]	Investigate differences in perceived social support, depressive symptoms, and well-being among students with physical disabilities living in general student housing and those living in a disabled community.	Quantitative results showed statistically significant differences between the two groups in all three measures (perceived social support, depressive symptoms, and well-being). Open-ended responses revealed that those living in the disabled community felt more connected with other physically disabled students. However, it was also observed that many students with physical disabilities felt socially disconnected for various reasons.
Petcu et al. (2021) [32]	Explore differences between characteristics and behaviours of students with ASD and students with learning disabilities.	The characteristics of these two groups of first-year university students were similar, except in terms of gender, ethnicity, being the first generation in their family to attend university, and parental income. In comparison to first-year students with learning disabilities, students with ASD were less likely to engage in risky behaviours and use health services and the writing center.

**Table 4 behavsci-14-00111-t004:** Group 4: Impact of the pandemic on university students with disabilities.

Authors	Objectives	Key Findings
Loginova et al. (2023) [12]	Identifying the characteristics of psychological well-being among college students with various somatic health conditions in pre-pandemic- and COVID-19-associated periods.	The scores on the “Autonomy”, “Environmental Mastery”, and “Personal Growth” scales for the latter group experienced a sharp decline, indicating an overall decline in psychological well-being.
Madaus et al. (2022) [13]	Examining the experiences and perceptions of students with disabilities (SWDs) during the academic year 2020–2021, following the rapid shift to remote education due to the COVID-19 pandemic.	Most elements related to access to services and instruction did not show improvements from the spring semester of 2020. Additionally, elements related to mental health, motivation for learning, and connections with peers were perceived as worse than in the spring of 2020.

## Data Availability

Not applicable.

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
