# Peer review of "Trends and Challenges in the Mental Health of University Students with Disabilities: A Systematic Review"

_behavsci, 2024, doi:10.3390/bs14020111_

Round 1

Reviewer 1 Report

Comments and Suggestions for Authors

 I would like to thank you for giving me this opportunity to evaluate this scientific paper, a systematic review that examines mental health in university students with disabilities, focusing on increasing prevalence and associated challenges.

This manuscript reports new findings and is theoretically based on the current literature.

Comments about the paper:

2.2.2. Information sources and search strategy

This systematic review focused on sourcing complete works from primary publica-124 tions available in the Web of Science (WOS), SCOPUS, and PsycINFO databases

Why haven’t the authors search on Pubmed?

One of the objectives was:

Analyze the prevalence and associated risk factors of mental health problems among university students with disabilities.

I consider that the authors didn’t analyze this objective, the data is presented in Table 1, but there is no data on prevalence of mental health problem.

In this part the authors should give some numbers and some definition from the papers (how the mental health status was evaluated).

The papers are published between 2018-2023, the authors should emphasize the differences before/during-after pandemic.

Author Response

Dear reviewer, the authors deeply appreciate the suggestions made, which will significantly contribute to improving the quality of the current article.

The following is a response to each of the questions and suggestions made, these have been marked in red in the article.

Review 1

Response

2.2.2. Information sources and search strategy

This systematic review focused on sourcing complete works from primary publica-124 tions available in the Web of Science (WOS), SCOPUS, and PsycINFO databases

Why haven’t the authors search on Pubmed?

We made this decision based on the following criteria: WOS and SCOPUS cover a wide range of disciplines, proving useful when seeking an interdisciplinary perspective or a broader view of research in a particular area. On the other hand, PsycINFO specializes in psychology and related disciplines, making it potentially more comprehensive and specific in our search.

While PubMed is a valuable source for biomedical and health literature, its scope is more limited compared to WOS and SCOPUS. As our research does not primarily focus on biomedical sciences, we opted for WOS, SCOPUS, or PsycINFO. We have included an explanation in the text.

Analyze the prevalence and associated risk factors of mental health problems among university students with disabilities.

I consider that the authors didn’t analyze this objective, the data is presented in Table 1, but there is no data on prevalence of mental health problem.

In this part the authors should give some numbers and some definition from the papers (how the mental health status was evaluated).

We appreciate this suggestion and understand that it was a shortcoming in our study. We have incorporated specific statistical data into the discussion.

The papers are published between 2018-2023, the authors should emphasize the differences before/during-after pandemic.

The last 5 years have been selected to obtain a current perspective on the subject.

We have included a brief explanation in section 3.1.

Reviewer 2 Report

Comments and Suggestions for Authors

Thank you for the opportunity to review your manuscript. I have a few questions: 1. Did you search by hand trying to find other articles? This could be considered a "study risk of bias assessment."2. Why did you decide on the time period of 2018-2023? 3. Were there only two reviewers collecting data? 4. What was the process used to resolve disagreements between data collectors? 5. Do you have a copy of the article by Page et al. (2021) titled, PRISMA 2020 explanation and elaboration: Updated guidance and exemplars for reporting systematic reviews.

Comments on the Quality of English Language

Please see the document attached. I only made corrections on pp. 1-2. 

On page 7, line 245, the word DATA is the plural form of DATUM. This means that the verb must be in the plural form -- "Data were...."

We usually use a comma before the word SUCH. Correction is needed throughout the document.

On page 8, line 285 and line 290: The adverb expression, "On the other hand," is used on both lines.  Other ways to say that: instead, conversely, then again, in contrast...

On page 5, line 187, use of Spanish instead of English.

On page 9, lines 322-325: The paragraph is written in Spanish.

Author Response

Dear reviewer, the authors deeply appreciate the suggestions made, which will significantly contribute to improving the quality of the current article.

The following is a response to each of the questions and suggestions made, these have been marked in red in the article.

Review 2

Response

 1. Did you search by hand trying to find other articles? This could be considered a "study risk of bias assessment."

We attempted to search for additional literature, including grey literature, on Google Scholar, among other sources. However, this search posed challenges in terms of data systematization, and therefore, we chose to focus solely on scientific literature with impact in indexed databases. While we are aware that many studies are significant and gain visibility even if they are not indexed, and though they may not be included in this review, we acknowledge their importance for our project.

2. Why did you decide on the time period of 2018-2023?

The last 5 years have been selected to obtain a current perspective on the subject.

We have included a brief explanation in section 3.1.

3. Were there only two reviewers collecting data?

After removing duplicate articles, the three researchers evaluated 340 articles to verify if they were relevant to the topic (i.e., addressing variables related to mental health or emotional well-being in university students with disabilities).

Subsequently, two of the researchers conducted an in-depth reading of the remaining 85 articles, with the third researcher overseeing the process to ensure a consensus decision in case of any disagreement regarding the relevance of an article.

We have included a brief explanation in section 2.2.3.

4. What was the process used to resolve disagreements between data collectors?

A review protocol with specific inclusion/exclusion criteria was established in the Parsifal tool to provide a clear framework that could be consulted when disagreements arise. In cases of disagreement, an independent review of the data was conducted by the third team member to impartially assess the discrepancy. Since there were hardly any discrepancies, it was not necessary to conduct an iterative review that would require multiple rounds of discussion and adjustments to reach the final consensus.

5. Do you have a copy of the article by Page et al. (2021) titled, PRISMA 2020 explanation and elaboration: Updated guidance and exemplars for reporting systematic reviews.

Yes, we have attempted to adhere to the PRISMA guidelines as much as possible, although we have not progressed to meta-analysis. Regarding the mentioned article, we checked its adherence to the essential elements as outlined in box 2. While it is true that we did not meet requirements 11 and 12 (explanation provided in lines 151-158), we appreciate the reviewer's insistence as we had not specifically included point 9 from that box, and it is an essential element. Therefore, it is now included in the study (lines 159-171). SINCERE THANKS.

On page 7, line 245, the word DATA is the plural form of DATUM. This means that the verb must be in the plural form -- "Data were...."

We usually use a comma before the word SUCH. Correction is needed throughout the document.

On page 8, line 285 and line 290: The adverb expression, "On the other hand," is used on both lines.  Other ways to say that: instead, conversely, then again, in contrast...

On page 5, line 187, use of Spanish instead of English.

On page 9, lines 322-325: The paragraph is written in Spanish.

Thank you for the level of detail; we have proceeded to make the changes.

Reviewer 3 Report

Comments and Suggestions for Authors

The current systematic review, based on PRISMA protocol approach, aimed at investigating the trends and Challenges in Mental Health of University Students with disabilities. The paper provides scientific contribution to the panorama of studies investigating on interesting issue focused on 1) the difficulties and perspectives about the academic education of students with disabilities and their training for future employment; 2) the need to provide for their physical and mental health. The topic is interesting enough, contributing to achieving the several Global Goals of the 2030 Agenda for Sustainable Development focused on equality, high and inclusive education and well-being. Today, despite significant and growing improvements in universities, students with disabilities encounter many difficulties. For this reason, the present study is appreciable because it focuses on a population whose participation in the academic context is increasing and which is leading universities to adopt adequate resources.

However, some changes and explanations to improve quality of manuscript for his acceptance should be done.

The topic is very broad and complex and for this reason, inaccuracies, gaps and the need for further investigation are highlighted below.

The clarity of the manuscript is good enough in the language style, syntax, and sentence construction; however, the text tends to be confusing and not very organic.

Abstract section

In the ‘abstract section’ the Authors should specify how many articles and how they were selected and included in the review.

Introduction section

The study starts from the observation that over time the educational system, even at higher levels of education, provides increasingly wider access to different populations including students with various disabilities such as intellectual, motor, sensory disabilities, autism spectrum disorders and of neurodevelopment etc, prefiguring an increasingly fair and inclusive educational system with resources, measures, programs and services aimed at encouraging the achievement of high levels of academic qualifications. In fact, higher institutions such as universities have the task of promoting the holistic development of students who have multiple health needs; Universities are called upon to play a crucial role in promoting the inclusion of all students including students with disabilities, thus requiring significant changes in educational systems to achieve equity of opportunity. In this way, many students who were previously excluded from the education system, especially higher education ones, can now aspire to find acceptance in those universities that are equipped to guarantee it.

All university students during their university career are subjected to high levels of stress which can lead to the onset of mental suffering with an increase in negative dimensions such as loss of hope, sense of loneliness or self-esteem. In particular, it is known that medical students suffer an increased risk of depression compared to their peers currently enrolled in non-medical university courses and the presence of depressive symptoms seems to occur as early as the 1st year of the students' medical education (listed below), especially in women.

In this context, as reported by Authors, the students with disabilities face unique and additional challenges in adapting and meeting the demands of university life.

There are few studies focused on the prevalence of emotional disorders among university students with disabilities compared to the student population without disabilities. The few studies demonstrate a higher rate of depression, suicide risk, self-harm, and anxiety in populations with disabilities. Studies on the factors that contribute to the emotional difficulties of students with disabilities, in addition to the limitations associated with it, have highlighted the role of stigma still widespread towards students with disabilities who feel discriminated against and not welcomed, therefore marginalized and excluded (linked below).

Recent study examines the need to integrate disability health education into medical school curricula and shares successful training examples that can serve as a framework for how to accomplish this. In fact, people with disabilities encounter significant health and health-care inequities yet disability health training in medical education remains inadequate. The authors could take it into account (linked below).

The Covid 19 Pandemic  has exacerbated the stressed conditions experienced by university students, even more so by the most vulnerable ones, who have had to face new challenges in a remote educational environment and in the context of uncertainty and isolation. Regarding the impact of recent Pandemic on the mental health of college students, it would be useful to mention, for example, recent international studies (Linked below).

At the end of introduction, the authors clearly describe the objectives of the study.

Methods and materials section

The Authors try to answer important questions on 1) the prevalence of mental disorders: 2) on the risk factors that increase the fragility of disabled students with increased distress: 3) on the difficulties and challenges they encounter due to being disabled ; 4) on barriers in accessing services and 5) on difficulties during the pandemic period. Whitin “Phenomenon of interest “there is no mention of university support services.

The tables are exhaustive for the accurate data presentation.

Discussion and conclusions section

The authors concluded  that there is a higher prevalence of mental health issues among students with disabilities compared to their non-disabled peers, especially about  cognitive disabilities.

The narrative method allows a good description of the existential difficulties of the disabled person but does not allow us to outline operational strategies and recommendations other than the generic and tautological ones with which the study concludes.

Furthermore, the study does not take into account the students with mental disorders such as schizophrenia or personality disorders as if students with mental disorders did not exist but it focuses only on emotional disorders (anxiety, depression, etc.); the authors do not focus on  academic performance that this population manages to achieve in terms of duration of the academic path, acquisition of knowledge, achievement of the qualification; no mention of the need for technological innovations at an educational level which are necessary to compensate for learning difficulties was carried out; the authors do not take into account the issue that disabled student population can enroll in degree courses qualifying for the healthcare profession which involve attending internships in healthcare setting where a relationship between professional and patient is required; the authors do not take into account the objective difficulties in acquiring skills and competences due to specific disabilities that prevent students with disabilities from carrying out certain professions; finally, post-graduation employment outcomes are not taken into account.

In relation to the last issue, unfortunately the post-graduation path of people with cognitive disabilities, neurodevelopmental disorders and mental disorders remains, to date, still an unsolved problem that still requires attention and efforts.

In addition, the authors should report how their study contributes to achieving the Global Goals of the 2030 Agenda for Sustainable Development.

Suggested references

https://pubmed.ncbi.nlm.nih.gov/34220610/

https://pubmed.ncbi.nlm.nih.gov/34526153/

https://pubmed.ncbi.nlm.nih.gov/36109318/

https://pubmed.ncbi.nlm.nih.gov/32504342/

https://www.ncbi.nlm.nih.gov/pmc/articles/PMC8718341/

Author Response

Dear reviewer, the authors deeply appreciate the suggestions made, which will significantly contribute to improving the quality of the current article.

The following is a response to each of the questions and suggestions made, these have been marked in red in the article.

Review 3

Response

In the ‘abstract section’ the Authors should specify how many articles and how they were selected and included in the review.

Other comments and suggestions

Totally agree, we have added the information to the abstract.

The authors highly appreciate the suggestions made by this reviewer. Following these suggestions, new references have been incorporated, and a cross-cutting perspective regarding the Global Goals of the 2030 Agenda for Sustainable Development has been included. Likewise, we agree with the reviewer regarding the scarcity of studies addressing the perspective of students with a pre-existing diagnosis of mental illness, and undoubtedly, this will be something we will take into account for future work.

Round 2

Reviewer 2 Report

Comments and Suggestions for Authors

The manuscript has improved.

Comments on the Quality of English Language

There are still grammatical problems. For example, the authors need to decide if they are going to capitalize or not the word AUTISM.  It should be uniform throughout the manuscript. The authors also use the expression "students in the spectrum and students with disabilities..." What spectrum? Students with autism are students with disability. There are a few punctuation problems. There are paragraphs with less than 3 complete sentences. There are places where the authors should have used e.g.,  (See p.1).

Author Response

Dear reviewer, we appreciate once again your thoroughness and efficiency. As you rightly pointed out, we have proceeded to standardize the text using the acronym ASD (previously stated), included the use of "e.g.," and reviewed punctuation aspects, in addition to correcting any typos found. Thank you for your dedication.